# Integrated Dynamic Characterization of Thermorheologically Simple Viscoelastic Materials Accounting for Frequency, Temperature, and Preload Effects

**DOI:** 10.3390/ma12121962

**Published:** 2019-06-18

**Authors:** Eduardo G. Olienick Filho, Eduardo M. O. Lopes, Carlos A. Bavastri

**Affiliations:** 1Mechanical Engineering Department, Technological Federal University of Parana, Curitiba 81280-340, Brazil; 2Mechanical Engineering Department, Federal University of Parana, Curitiba 81530-900, Brazil; eduardo_lopes@ufpr.br (E.M.O.L.); bavastri@ufpr.br (C.A.B.)

**Keywords:** fractional derivatives, integrated dynamic characterization, hybrid optimization, viscoelastic materials

## Abstract

In vibration insulation projects, a parameter affecting the dynamic properties of the viscoelastic materials is the previous static load acting on the supports, denominated here as the ‘preload’. Most of the currently-used methodologies obtain the dynamic properties by considering only the effects of temperature and frequency. The additional effect of preload can be added to the usual methodologies by employing the hyperelastic theory developed by Mooney–Rivlin. The current work proposes an integrated approach to characterize thermorheologically simple viscoelastic materials, including the preload effect along with the influence of temperature and frequency. The proposed method uses a hybrid optimization process, combining a genetic algorithm (GA) and a non-linear optimization technique—named ‘simplex’—in an inverse problem structure applied to all experimental data at hand. A set of samples of elastomer BT-806 55 (butyl rubber) was tested at various temperatures, frequencies, and preloads. The comparison between the results of the present methodology and traditional approaches to a variation in the dynamic properties at all frequencies and temperatures for a constant vibration amplitude. The present results prove that the proposed methodology is a viable alternative to represent the dynamic properties of materials used in vibration isolation.

## 1. Introduction

In vibration control, the use of viscoelastic materials (VEMs) is becoming more and more frequent. In order to make this use feasible and efficacious, knowledge on their dynamic properties is fundamental. Researchers [1,2,3,4,5,6] have devoted many efforts to develop a reliable methodology to accurately represent the dynamic behavior of viscoelastic materials, especially by the use of fractional calculus.

The beginning of modern applications of fractional calculus in linear viscoelasticity is attributed to Bagley [1]. After this work, Bagley and Torvik presented several papers in this line of research [2,3,4,5]. 

By expressing the pertinent constitutive relations using fractional derivatives, it is possible to characterize a thermorheologically simple viscoelastic material using fewer parameters. In particular, using the four-parameter Zener model [7], this type of material can be represented with a fairly good degree of accuracy.

The transformation of the governing relations of relaxation and fluency in time domain into frequency domain relationships [8] can be performed through the methodology presented by Birk and Song [9], which permits a suitable characterization in the frequency domain.

An experimental technique for a complete dynamic characterization of a viscoelastic material is presented by Caracciolo et al [10] through the determination of Poisson’s ratio and Young’s complex modulus. The technique uses a beam that is excited by an electromagnetic shaker. Experimental curves are obtained for both properties and then merged into a single curve by using the variable-reduction method.

Zhao et al [11] presented an explicit semi-analytical numerical model and a numerical model based on the finite-difference theory—in order to find the dynamic characteristics of homogeneous and locally-non-homogeneous viscoelastic materials. The inverse identification problem, using a two-dimensional approach for a formulation via Kelvin–Voigt fractional model, is solved by using an optimization technique.

A mathematical model already established in representing the dynamic behavior of VEMs is based on Zener’s mechanical model. In the constitutive equation, the fractional calculus introduces great advantages over the use of whole-order derivatives. The capacity of considering memory in the behavior of the material allows the use of a smaller number of parameters in the assembly of the mathematical model.

Temperature has the most important influence on the behavior of VEMs and its effect can be introduced by means of an empirical equation known as WLF (Williams–Landel–Ferry) equation [12,13], which is obtained from the frequency–temperature superposition principle. In practical terms, the effect of temperature is reflected in the results as a shift of the curves of dynamic modulus and of the loss factor in relation to frequency. This can be conveniently presented by a plot termed ‘reduced frequency nomogram’ [14], which is constructed from the consolidation of the curves of dynamic properties.

The identification of the mathematical model parameters regarding the description of the viscoelastic behavior by fractional derivatives with four parameters requires two previous and distinct steps. In the first, tests are carried out to measure the desired property—complex Young’s modulus or complex shear modulus—under pre-established frequency and temperature conditions. In the second step, several dynamic curves, as functions of temperature and frequency, are plotted and the superposition principle is applied in order to collapse them all at a reference temperature, generating a pair of master curves. Those master curves are displayed in the reduced frequency nomogram [15].

A limitation regarding the currently used nomograms is that they do not consider effects that can be relevant in vibration isolation designs. Viscoelastic bases of machinery exposed to vibrations are subject to significant static load effects (own weight) that may produce changes in the dynamic properties of viscoelastic materials.

Lopes et al [16] presented a new approach for the dynamic characterization, in which all the experimental data are dealt with a single time. The method, called ‘integrated dynamic characterization’ is based on a curve fitting procedure involving numerical and experimental transmissibility curves over a certain frequency band and at different temperatures.

An approach to the characterization of a viscoelastic material via generalized (fractional) derivatives was presented by Boiko et al [17]. It also allows the identification of dynamic properties through the transmissibility functions in wide ranges of frequency and temperature. An annular specimen submitted to forced vibration is employed in order to obtain the complex modulus of elasticity and the Poisson’s ratio.

Zopf et al [18] presented two approaches regarding the use of fractional derivatives based on the Zener solid model, considering large deformations. In both approaches, the Zener model is composed of a nonlinear spring and a fractional Maxwell element. The model was developed under considerations of finite deformation theory. The computational algorithm for the fractional derivative in time is the difference between the two approaches.

Bonfiglio et al [19] introduced a new method for the determination of complex Young’s modulus values for isotropic and homogeneous VEMs in the frequency domain. The method can be applied for simulations and optimizations regarding the insertion of a VEM in vibration and/or sound insulation applications.

Sousa et al [20] presented a method to determine the parameters of a fractional model for VEMs, considering the effects of frequency and temperature. The data were obtained by sampling a nomogram supplied by the manufacturer, using at least two different temperatures. Through a hybrid optimization procedure, the constitutive model parameters were determined.

In the present work, the preload effect is added to the mathematical modeling of the desired dynamic property. It is important to consider that, in such a situation, two types of loadings are present: (I) The static loading (preload), usually with constant value and (II) the dynamic loading due to the vibration amplitude which can often be variable. In [13], it was revealed that the effects of static and dynamic loads are weaker than that of temperature, and that they operate differently in the material. 

The static loading (preload) presents itself as a nonlinear behavior, according to [13]. The value of the dynamic modulus goes up with preload increase whereas the value of the loss-factor decreases. The solution to consider the influence of preload in the problem of dynamic characterization of a VEM is carried out through the Mooney–Rivlin equation with specific considerations to the case [13,21,22].

The dynamic loading tends to induce a behavior similar to that of temperature and, in this case, a superposition principle similar to that variable may be applied. A general nonlinear behavior may be observed for large values of dynamic displacements in pure VEMs. VEMs with the addition of carbon black or silica may present nonlinearity already at low amplitudes [23]. 

In order to adapt the fractional derivative model to add the preload effect, a constant dynamic displacement value is considered in the present work. The formulation of the problem is complemented by the simultaneous inclusion of the effects of temperature and preload into the fractional model, in which the effect of frequency is already present. To the authors’ knowledge, this integrated approach has not been presented so far.

The identification of the parameters of the proposed model is performed along with the identification of the parameters of the influence factors of temperature and preload. That occurs in a simultaneous fashion from the experimental data in the frequency domain. This identification process is carried out through a computational program, developed in MATLAB, using a hybrid optimization technique combining a genetic algorithm (GA) and the simplex method developed by Nelder and Mead [24] in an inverse problem structure. This method was adapted from ‘integrated dynamic characterization’ [16] method.

## 2. Mathematical Methods 

The models presented in the current work are directed to isotropic VEMs with simple thermorheological behavior. They are presented as follows.

### 2.1. Dynamic Characterization—A Fractional Derivative Approach

The fractional calculus applied to Zener’s mechanical model is a particular case of the general constitutive equation named ‘operator equation’ [25]. It has been successfully used for describing the linear viscoelastic behavior of polymers [6].

Bagley [1] presented a theoretical development for the application of fractional calculus in the representation of viscoelastic behavior. This contribution was later followed by other papers [2,3,4,5]. The four-parameter fractional model allows an adequate representation to describe the dynamic behavior of thermorheologically simple viscous materials with a reduced number of parameters, complying with the second law of thermodynamics [26]. 

Two definitions of fractional derivatives can be applied aiming at using it in the constitutive equation of the VEMs [25]: The one by Riemann–Liouville and the one by Caputo. Both fractional derivative approaches are equivalent in the representation of viscoelastic behavior, although Mainardi [25] attributes to Caputo’s derivative the primacy of that representation, considering that the latter is fully compatible with the classical approach, with entire derivatives. 

With β being the fractional derivative order, *σ*(*t*) the stress, and *ε*(*t*) the strain, both in time domain, the constitutive equation for Zener’s model, using fractional derivatives, is as follows:(1)$$\sigma \left(t\right)+b\frac{d^{\beta }\sigma \left(t\right)}{dt^{\beta }}=E_{o}\varepsilon \left(t\right)+E_{1}\frac{d^{\beta }\varepsilon \left(t\right)}{dt^{\alpha }}.$$

Considering that
(2)$$\bar{F}\left(\Omega \right)=ℑ\left\{D^{\beta }\left[f\left(t\right)\right]\right\}={\left(i.\Omega \right)}^{\beta }ℑ\left[f\left(t\right)\right].$$

Equation (1) may be rewritten in the frequency domain as follows:(3)$$\bar{\sigma }\left(\Omega \right)+b{\left(\Omega .i\right)}^{\beta }\bar{\sigma }\left(\Omega \right)=E_{o}\bar{\varepsilon }\left(\Omega \right)+E_{1}{\left(\Omega .i\right)}^{\beta }\bar{\varepsilon }\left(\Omega \right),$$where the overbar indicates that the magnitude is complex, and $i=\sqrt{-1}$. Manipulating Equation (3), one obtains that
(4)$$\frac{\bar{\sigma }\left(\Omega \right)}{\bar{\varepsilon }\left(\Omega \right)}=\bar{E}\left(\Omega \right)=\frac{E_{o}+E_{1}{\left(\Omega .i\right)}^{\beta }}{1+b{\left(\Omega .i\right)}^{\beta }},\;\mathrm{with}\;0<\beta <1.$$

The $\bar{E}\left(\Omega \right)$ in Equation (4) is the complex Young’s modulus, and may be written as
(5)$$\bar{E}\left(\Omega \right)=E\left(\Omega \right)+i.E^{\prime }\left(\Omega \right),$$
where *E*(Ω) is the real part of the complex modulus, called ‘dynamic modulus’ and representing the elastic component; and *E’*(Ω) is the imaginary part, representing the dissipative component, both in the frequency domain. 

Equation (5) can be further represented by
(6)$$\bar{E}\left(\Omega \right)=E\left(\Omega \right)\left[1+i.\eta \left(\Omega \right)\right].$$
where
(7)$$\eta \left(\Omega \right)=\frac{E^{\prime }\left(\Omega \right)}{E\left(\Omega \right)},$$
is called ‘loss factor’.

Setting $E_{1}=E_{\infty }.b$, in Equation (4), one obtains that
(8)$$\bar{E}\left(\Omega \right)=\frac{E_{0}+E_{\infty }b{\left(\Omega .i\right)}^{\beta }}{1+b{\left(\Omega .i\right)}^{\beta }},$$
where $E_{0}$ is the elastic modulus, $E_{\infty }$ is the vitreous modulus, b is a real parameter associated to the relaxation time, and β, has defined before, is the fractional derivative order, which varies between 0 and 1. The same considerations made for complex Young’s modulus may be made for the $\bar{G}\left(\Omega \right)$—complex shear modulus.

The materials represented by Equation (8) are usually polymers of the elastomeric type and have their dynamic behavior strongly influenced by external factors, such as temperature and dynamic/static loads, among other parameters. However, temperature is the only external factor usually considered in vibration control design.

### 2.2. Effect of Temperature

The difference between the effects of temperature and frequency is that they are qualitatively inverse, but the order of magnitude of both effects is very different. According to [14], regarding typical elastomers, the difference between the effects of temperature and frequency is very significant. The reciprocity condition allows to obtain a comprehensive characterization from limited experimental data through the so-called ‘frequency-temperature superposition principle’ [12].

This principle establishes that various curves of dynamic properties can be superimposed on a chosen reference temperature, by means of frequency displacements, forming two single master curves for the dynamic modulus and loss factor. To that end, it is necessary to establish a displacement factor that permits realigning the aforementioned curves [13,15].

An expression that translates such effect very well is the Williams–Landel–Ferry (WLF) model [12], given by
(9)$$\log\alpha_{T}\left(T\right)=\frac{-\theta_{1}\left(T-T_{0}\right)}{\theta_{2}+T-T_{0}},$$
where $\log\alpha_{T}$ is the decimal base logarithm of $\alpha_{T}$, the displacement factor due to temperature, *θ*_1_ and *θ*_2_ are parameters to be determined for each material, *T* is the temperature at which experimental data are collected, and *T*_0_ is the reference temperature.

Substituting Equation (9) in Equation (8) leads to the following equation:(10)$$\bar{E}\left(\Omega ,T\right)=\frac{E_{0}+E_{\infty }b{\left[\alpha_{T}\left(T\right).\Omega .i\right]}^{\beta }}{1+b{\left[\alpha_{T}\left(T\right).\Omega .i\right]}^{\beta }}.$$

### 2.3. Effect of Static Load 

In some circumstances, the static load is a significant factor, impossible to be disregarded in vibration control designs. This is particularly true in vibration isolation actions.

A mathematical model to predict the behavior of a hyperelastic material is given in [21,22]. The resulting equation is known as the ‘Mooney–Rivlin equation’. This theory was used in [13] to implement the effect of the static load in the model presented in Section 2.1 for the identification of viscoelastic materials.

The Mooney–Rivlin equation is used to describe the stress-strain relationships for values of extensional strains (λ) varying up to 3 [13]. In the Mooney–Rivlin model, the strain energy density function W for an incompressible material, said *Mooney–Rivlin material* [21,22] is given by
(11)$$W=C_{1}\left({\bar{I}}_{1}-3\right)+C_{2}\left({\bar{I}}_{2}-3\right),$$
where $C_{1}$ and $C_{2}$ are experimentally determined constants of the material, and ${\bar{I}}_{1}$ and ${\bar{I}}_{2}$ are strain invariants defined as
(12)$${\bar{I}}_{1}=tr\left(B\right),$$
(13)$${\bar{I}}_{2}=\frac{1}{2}\left[I_{1}^{2}-tr\left(B\right)\right],$$
and **B** is the left Cauchy–Green strain tensor, and *tr* (**B**) is the trace of **B.** The **B** tensor is given by
(14)$$B=FF^{T},$$
where **F** is the strain gradient, and **F***^T^* represents the transpose of the strain gradient.

For incompressible hyperelastic materials, a third invariant (${\bar{I}}_{3}$) is constant and equal to 1 [det (**B**) = 1]. This is what makes the energy function dependent only on the first two invariants. Thus, for an incompressible material, the Mooney–Rivlin expression, Equation (11), becomes
(15)$$W=C_{1}\left(\lambda_{1}{}^{2}+\lambda_{2}{}^{2}+\lambda_{3}{}^{2}-3\right)+C_{2}\left(\lambda_{1}{}^{2}\lambda_{2}{}^{2}+\lambda_{2}{}^{2}\lambda_{3}{}^{2}+\lambda_{3}{}^{2}\lambda_{1}{}^{2}-3\right),$$
with $\lambda_{1}\lambda_{2}\lambda_{3}=1$, where $\lambda_{1}$, $\lambda_{2}$, and $\lambda_{3}$ represent the extensional strains regarding the main (engineering) strains, as λ=ε+1.

According to [21,22], the components of Cauchy’s stresses, for a hyperelastic material, are given by
(16)$$\sigma_{11}-\sigma_{33}=\lambda_{1}\frac{\partial W}{\partial \lambda_{1}}-\lambda_{3}\frac{\partial W}{\partial \lambda_{3}},$$
(17)$$\sigma_{22}-\sigma_{33}=\lambda_{2}\frac{\partial W}{\partial \lambda_{2}}-\lambda_{3}\frac{\partial W}{\partial \lambda_{3}},$$

Using Equations (15)–(17) with $\lambda_{1}\lambda_{2}\lambda_{3}=1$, one obtains
(18)$$\sigma_{11}-\sigma_{33}=2C_{1}\left(\lambda_{1}{}^{2}-\lambda_{3}{}^{2}\right)-2C_{2}\left(\frac{1}{\lambda_{1}{}^{2}}-\frac{1}{\lambda_{3}{}^{2}}\right),$$
(19)$$\sigma_{22}-\sigma_{33}=2C_{1}\left(\lambda_{2}{}^{2}-\lambda_{3}{}^{2}\right)-2C_{2}\left(\frac{1}{\lambda_{2}{}^{2}}-\frac{1}{\lambda_{3}{}^{2}}\right),$$
(20)$$\sigma_{11}-\sigma_{33}=2C_{1}\left(\lambda^{2}-\frac{1}{\lambda }\right)-2C_{2}\left(\frac{1}{\lambda^{2}}-\lambda \right),$$
or $\sigma_{22}=\sigma_{33}=0$, $\lambda_{1}=\lambda ,\;\lambda_{2}=\lambda_{3}=\frac{1}{\sqrt{\lambda }}$, with $\sigma_{22}-\sigma_{33}=0$, and
(21)$$\sigma_{11}=\sigma =\left(2C_{1}+\frac{2C_{2}}{\lambda }\right)\left(\lambda^{2}-\frac{1}{\lambda }\right),$$
it follows that
(22)$$\sigma =2\left(C_{1}\lambda +C_{2}\right)\left(\lambda -\frac{1}{\lambda^{2}}\right).$$

Considering the relationship between the extensional strain and the engineering strain, that is, λ=ε+1, in Equation (22), it results that
(23)$$\sigma =2\left[C_{1}\left(\varepsilon +1\right)+C_{2}\right]\;\left[\frac{\varepsilon \left(\varepsilon^{2}+3\varepsilon +3\right)}{{\left(\varepsilon +1\right)}^{2}}\right],$$
which may be written as
(24)$$\frac{\sigma }{\varepsilon }=E_{o}=2\left[C_{1}\left(\varepsilon +1\right)+C_{2}\right]\;\left[\frac{\left(\varepsilon^{2}+3\varepsilon +3\right)}{{\left(\varepsilon +1\right)}^{2}}\right].$$

The implementation of the combined effects of static and dynamic loads (the latter resulting from the vibration excitation) will be performed considering that stress is composed of a component relative to the preload and another component related to the dynamic load, in such a way that
(25)σ(Ω,λ)=F(λ).Q(Ω).

In Equation (25), F(λ) is the component of stress σ(Ω,λ) due to preload, and Q(Ω) is a component due to the frequency or dynamic load. The component due to preload in Equation (25) may be replaced by Equation (22), which gives
(26)$$\sigma \left(\Omega ,\lambda \right)=2\left(C_{1}\lambda +C_{2}\right)\left(\lambda -\frac{1}{\lambda^{2}}\right)Q\left(\Omega \right).$$

Considering Hooke’s law, one can get to Equation (27), which can be used to describe the combined properties of the static and dynamic modulus of elastomers. That is
(27)$$d\sigma \left(\Omega ,\lambda \right)=E\left(\Omega ,\lambda \right).\frac{d\lambda }{\lambda },$$
and
(28)$$E\left(\Omega ,\lambda \right)=\lambda .\frac{d\sigma \left(\Omega ,\lambda \right)}{d\lambda }.$$

Using Equation (26) in Equation (28), one obtains
(29)$$E\left(\Omega ,\lambda \right)=\left[2.C_{1}\left(\lambda^{2}-\frac{1}{\lambda }\right)+2\left(C_{1}\lambda +C_{2}\right)\left(\lambda +\frac{2}{\lambda^{2}}\right)\right]Q\left(\Omega \right),$$
or
(30)$$E\left(\Omega ,\lambda \right)=\left[C_{1}\left(4\lambda^{2}+\frac{2}{\lambda }\right)+C_{2}\left(2\lambda +\frac{4}{\lambda^{2}}\right)\right]Q\left(\Omega \right).$$

Equation (30) may be rewritten as
(31)$$E\left(\Omega ,\lambda \right)=\left[C_{1}F_{1}\left(\lambda \right)+C_{2}F_{2}\left(\lambda \right)\right]Q\left(\Omega \right),$$
where
(32)$$F_{1}\left(\lambda \right)=2\left[2\lambda^{2}+\frac{1}{\lambda }\right],$$
(33)$$F_{2}\left(\lambda \right)=2\left[\lambda +\frac{2}{\lambda^{2}}\right],$$

Using the complex modulus approach for VEMs [see Equations (5)–(7)], Equation (31) may be written as
(34)$$E\left(\Omega ,\lambda \right)\left[1+i.\eta \left(\Omega ,\lambda \right)\right]=\left\{C_{1}\left[1+i\eta_{1}\left(\Omega \right)\right]F_{1}\left(\lambda \right)+C_{2}\left[1+i.\eta_{2}\left(\Omega \right)\right]F_{2}\left(\lambda \right)\right\}Q\left(\Omega \right).$$

From the real and imaginary parts of Equation (34), one can extract the dynamic modulus and the loss factor, which are then given by
(35)$$E\left(\Omega ,\lambda \right)=\left[C_{1}F_{1}\left(\lambda \right)+C_{2}F_{2}\left(\lambda \right)\right]Q\left(\Omega \right),$$
and
(36)$$\eta \left(\Omega ,\lambda \right)=\left[C_{1}F_{1}\left(\lambda \right)\eta_{1}+C_{2}F_{2}\left(\lambda \right)\eta_{2}\right]Q\left(\Omega \right).$$

Considering that ε is very small and, therefore, λ→1, with $\eta_{1}$ assumed to be zero [12], the constant $\eta_{2}$ and the function Q(Ω) can be obtained. This will allow the following expressions be written from Equations (32)–(36), According to [13], this allows the following expressions to be written from Equations (32)–(36):(37)$$E\left(\Omega ,\lambda \right)=\left[C_{1}F_{1}\left(\lambda \right)+C_{2}F_{2}\left(\lambda \right)\right]\frac{E\left(\Omega \right)}{6\left(C_{1}+C_{2}\right)},$$
(38)$$\eta \left(\Omega ,\lambda \right)=\frac{\left(C_{1}+C_{2}\right)F_{2}\left(\lambda \right)}{C_{1}F_{1}\left(\lambda \right)+C_{2}F_{2}\left(\lambda \right)}\eta \left(\Omega \right).$$

Considering now the influence of temperature, Equations (37) and (38) may be expanded as follows: (39)$$E\left(\Omega ,T,\lambda \right)=\left[C_{1}F_{1}\left(\lambda \right)+C_{2}F_{2}\left(\lambda \right)\right]\frac{E\left(\Omega ,T\right)}{6\left(C_{1}+C_{2}\right)},$$
(40)$$\eta \left(\Omega ,T,\lambda \right)=\frac{\left(C_{1}+C_{2}\right)F_{2}\left(\lambda \right)}{C_{1}F_{1}\left(\lambda \right)+C_{2}F_{2}\left(\lambda \right)}\eta \left(\Omega ,T\right).$$

An important observation must be made regarding Mooney–Rivlin’s constants $C_{1}$ and $C_{2}$. In this application, for the optimization process, the $\frac{C_{2}}{C_{1}}$ relationship shows to be the most adequate. Considering the relationship $\frac{C_{2}}{C_{1}}=k_{pc}$, Equations (39) and (40) become
(41)$$E\left(\Omega ,T,\lambda \right)=\left[C_{1}F_{1}\left(\lambda \right)+k_{pc}.C_{1}F_{2}\left(\lambda \right)\right]\frac{E\left(\Omega ,T\right)}{6\left(C_{1}+k_{pc}.C_{1}\right)},$$
and
(42)$$\eta \left(\Omega ,T,\lambda \right)=\frac{\left(C_{1}+k_{pc}.C_{1}\right)F_{2}\left(\lambda \right)}{C_{1}F_{1}\left(\lambda \right)+k_{pc}.C_{1}F_{2}\left(\lambda \right)}\eta \left(\Omega ,T\right).$$

The introduction of the factor $k_{pc}$ allows to eliminate $C_{1}$ and $C_{2}$ from equations (41) and (42). Simplifying Equations (41) and (42), it results that
(43)$$E\left(\Omega ,T,\lambda \right)=\left[F_{1}\left(\lambda \right)+k_{pc}.F_{2}\left(\lambda \right)\right]\frac{E\left(\Omega ,T\right)}{6\left(1+k_{pc}\right)},$$
and
(44)$$\eta \left(\Omega ,T,\lambda \right)=\frac{\left(1+k_{pc}\right)F_{2}\left(\lambda \right)}{F_{1}\left(\lambda \right)+k_{pc}.F_{2}\left(\lambda \right)}\eta \left(\Omega ,T\right).$$

Equations (43) and (44) are essentially the same as those presented in [13], as they have the same theoretical foundations. However, they have been adapted to fit in the context of the present study. 

## 3. Methodology

### 3.1. Material

The material used in the experimental tests for the present work is a copolymer of isobutylene (98%) and isoprene known as ‘butyl rubber’. This material has a good flexural behavior and a high damping capacity, the reason why it is largely used in devices intended to reduce or control vibrations. It is manufactured by Croslin Ltd. (São Paulo, Brazil) and referred to as BT-806 55.

### 3.2. Experiments

The butyl rubber (BT-806 55) was supplied in its raw state by the manufacturer, and the specimens were obtained after vulcanization at 160 °C for a period of 16.3 min. The rheometric curve was determined according to the ASTM D5289-12 Standard [27]. Forty-two specimens were obtained (for tests at 6 temperatures and 7 preload values), corresponding to an equivalent number of predetermined static strains. 

The tests were carried out in a polymer testing machine, MTS 831.50 (MTS Systems Corp, Eden Prairie, MN, USA), shown in Figure 1a, according to procedures established by the ASTM D5992-96 Standard [28]. Temperature conditions were ensured using a chamber with controlled temperature, shown in Figure 1b, in which the specimen was kept. The excitation frequency was also controlled throughout the tests and ranged from 1 to 1000 Hz.

The experimental data for the complex Young’s modulus were obtained according to conditions set forth in Table 1.

In order to obtain results that allow the convergence to a single solution for the optimization problem, the chosen temperatures must contemplate three characteristic regions of the VEMs under study: The vitreous, the transition, and the rubbery regions [13]. The value of the static strain due to preload, on which a constant excitation amplitude of +/−0.01 mm is applied, is named ‘average/medium strain’.

### 3.3. Integrated Identification 

The solution to the problem of identifying the parameters of the fractional derivative model and the other parameters describing effects that compose the models representing the influences of temperature and preload is obtained from an inverse problem using the experimental curves of the complex modulus for curve-fitting via a hybrid optimization process. The hybrid optimization is performed in two stages: The first, through a genetic algorithm (GA), allows to set initial values of unknown material parameters used in the second, which is a nonlinear process of optimization, based on the Nelder–Mead algorithm (fminsearch) [25] and implemented in MATLAB.

The standard optimization problem is defined by the search for a minimum of an objective function, which in the present context, is defined by the following equation: (45)$$f(x)=\sum_{k=1}^{r}\left\{\sum_{j=1}^{q}\left[\sum_{i=1}^{p}{\left(e_{ijk}\right)}^{C}\left(e_{ijk}\right)\right]\right\},$$
where superscript *C* represents the conjugate and $e_{ijk}$ is defined by
(46)$$e_{ijk}={\bar{E}}_{EXP}\left(\Omega_{i},T_{j},\lambda_{k}\right)-{\bar{E}}_{EST}\left(\Omega_{i},T_{j},\lambda_{k}\right),$$
with *i* = 1..*p*, *j* = 1..*q*, *k* = 1..*r*, where *p* is the number of excitation frequencies, *q* is the number of temperatures, and r is the number of preloads, while (${\bar{E}}_{EXP}$) represents the experimental value of the complex modulus and (${\bar{E}}_{EST}$) represents the estimated value for the same modulus, given by Equations (43) and (44) joined in Equation (6). The solution to the optimization problem will allow having the desired knowledge of all the parameters of the models, given by the following design vector:(47)$$x^{T}=\left[E_{o},\;E_{\infty },\;b,\;\beta ,\;T_{O},\;\theta_{1},\theta_{2},\;k_{pc}\right].$$

The computational cost of the optimization process may be relatively high, when the process involves the use of GA, as originally proposed in this work. When an initial estimate for the values of the parameters of Equation (47) is available, either by the use of a previously characterized VEM or by reference data available in the literature, the optimization process may become much faster. This can be done by implementing a by-pass in the GA method, starting directly with the simplex method (fminsearch). The gradient method can also be applied directly (by using the fmincon function of MATLAB, (2014a, The Mathworks Inc., Natick, MA, USA), although this requires that lower and upper limits are established for the parameter values. In this work, the identification procedure involved the use of the GA and fminsearch methods and bounds on the model parameters were not considered.

## 4. Results

For better quality of results, it was performed a pre-filtering of the experimental data in order to deal with problems such as excessive dispersion, physically incompatible isolated values, and exceptionally high complex modulus for the temperature range of concern. Therefore, to maintain a viable experimental data matrix for numerical processing, static strain values of 1% and 3%, in addition to frequencies above 500 Hz, were excluded from the analysis.

The options of the GA method were: Population size of 10,000 individuals, number of generations equal to 100, stop tolerance equal to 10^−11^ and mutation rate of 0.09. For the simplex optimization process, the following parameters were used: Maximal number of evaluations of the function equal to 5000, limit number of iterations equal to 600, and the convergence limit equal to $10^{-5}$ for the objective function *f*(*x*).

The numerical results obtained after the optimization process are the following: $E_{0}$ = 6.1306 MPa, $E_{\infty }$ = 496.57 MPa, b = 0.0020, β = 0.3166, $T_{0}=$ 293 K, $\theta_{1}=$ 7.1119, $\theta_{2}$ = 132.72, and $k_{pc}$ = −2.1930. These values correspond to the design vector of Equation (47). It is observed that the constants $C_{1}$ and $C_{2}$ of the Mooney–Rivlin model, the ratio of which is equal to $k_{pc}$, are relative to the material and their values are obtained from experimental data. However, in the current optimization procedure, the concern lies in their ratio, $k_{pc}$, and not in their separate values.

In general, the results can be evaluated using three types of plot:

I—a plot of the modulus of complex Young’s modulus ($\left|\bar{E}\right|$) versus frequency, for each experimentally tested condition of temperature and preload, comparing the curve regenerated (adjusted) with the parameters found in the integrated identification to the curve of experimental data, as shown in Figure 2 and Figure 3. It should be noted that λ=ε+1.

II—a plot of the loss factor versus the real part of the complex Young’s modulus, called Wicket plot. For a thermorheologically simple material, the complex modulus data points should cluster around a unique inverted-U shaped curve in a Wicket plot [30]. A viscoelastic material is regarded as thermorheologically simple if the frequency-temperature superposition principle holds, that is, if the frequency and temperature dependencies can be worked out in such a way that a single dependence on a compound variable, combining the effects of both frequency and temperature, results [13]. This compound variable is known as reduced frequency. Figure 4 is a typical portray of the Wicket plots for the five preload levels.

III—Reduced-frequency nomogram: Two extreme conditions of preload are presented in the same diagram, as shown in Figure 5. This is not usually given in these nomograms and constitutes an essential piece of information for vibration control designs, where heavy machinery may be supported by flexible elastomeric devices.

When the influence of preload was considered, the proposed parameter identification methodology performed satisfactorily, adequately reproducing the behavior of the investigated material. 

## 5. Discussion

According to the results obtained, using the same optimization methodology, a solution vector that does not consider the lambda value can be easily obtained, for the same set of experimental data. In this way the values of the dynamic modulus and loss factor can be compared for the same frequency and temperature. The error caused by not considering preload conditions can be obtained. When there is a 15% compressive strain preload, under conditions of constant vibration excitation in time of +/−0.01 mm, may reach up to 49% for the dynamic modulus and 23% for the loss factor, for the butyl rubber sample, under conditions of a constant temperature of 20 °C and 100 Hz [29]. This may lead to errors of up to 70% in the natural frequency of a single degree of freedom model, as computed according to [31]. 

Reduced frequency nomograms are very important in vibration isolation designs. They are usually supplied by manufacturers of viscoelastic materials employed in those applications and their classical form includes the representation of frequency and temperature effects. The use of the reduced frequency concept allows the extension of the experimental data to a larger frequency range and the temperatures are associated to isothermal lines in the nomograms.

Figure 6 shows a classical reduced frequency nomogram for butyl rubber BT806-55 in which 14 temperatures, ranging from 223 to 353 K, were included. By comparing the results of Figure 5 and Figure 6, it is possible to respectively establish a distinction between the results obtained in this work and in the traditional way.

The curves correspondent to dynamic modulus and loss factor to ε = 0, in Figure 5 and Figure 6, are equivalent, obtained in different test. Another curve of dynamic modulus and loss factor, in Figure 5, corresponding to ε = −0.15 of preload. The other two curves in Figure 5 represent the experimental results at strain ε = −0.15 (15% of compression). It is stressed that the values compared in both the cases correspond to the extremes of the test data.

It is important to observe that the experimental data employed to build the nomogram of Figure 6 are not the same as those employed in Figure 5. Another important feature is that a nonlinear behavior can be expected from elastomer samples at 15% of strain. It is also usual in practice that different samples supply different experimental results for the same viscoelastic material. In Figure 5, it can be observed that, due to the use of logarithmic scale, the curves are very close. If all the curves associated to the distinct preload values included in the experiments were plotted in the same nomogram, the reading of the dynamic properties would be considerably difficult.

That justifies the decision of building separate nomograms for each strain (preload) value. Therefore 5 different nomograms were built in this work, one for each preload value, as displayed in Figure A1, Figure A2, Figure A3, Figure A4 and Figure A5, Appendix A.

A comparison can be made between the ways of interpreting the nomograms included in the Appendix A and the traditional nomogram of Figure 6 (which includes the effects of frequency and temperature only). In the traditional nomogram, the design frequency (read at the right-hand vertical axis) and the temperature of concern (read at one of the inclined isothermal lines, or between them, if it is the case) allow the corresponding dynamic properties being obtained by the intersection of straight lines with the curves of dynamic modulus and loss factor.

When a given preload value, or strain due to this preload, is to be considered, the procedure is mostly the same, except for the case in which the preload value of concern is not available in any of the supplied nomograms. In this case, the values of the dynamic properties can be obtained by interpolating between the corresponding values for the same frequency and temperature extracted from two plots with strains closest to the desired value.

## 6. Conclusions

A methodology for an integrated dynamic characterization of thermorheologically simple viscoelastic materials was developed and implemented covering simultaneously the effects of temperature, frequency, and static loading in the form of preload.

The mathematical models and the optimization strategy resulted in computationally light approach, raising good prospects for the future inclusion of other effects besides those already considered in the dynamic characterization of viscoelastic materials, such as strain due to the dynamic excitation amplitudes. 

The proposed methodology is regarded as very promising, and its application in the generation of enhanced reduced-frequency nomograms can provide—for vibration isolator designs—more accurate information on the dynamic Young’s modulus when the viscoelastic isolators are exposed to preloads, which is not the case of nomograms currently supplied by VEM manufacturers. 

The effect of the preload could be presented in a single nomogram with the different curves for different preload values, or in several nomograms, one for each preload value.

## Figures and Tables

**Figure 1 materials-12-01962-f001:**
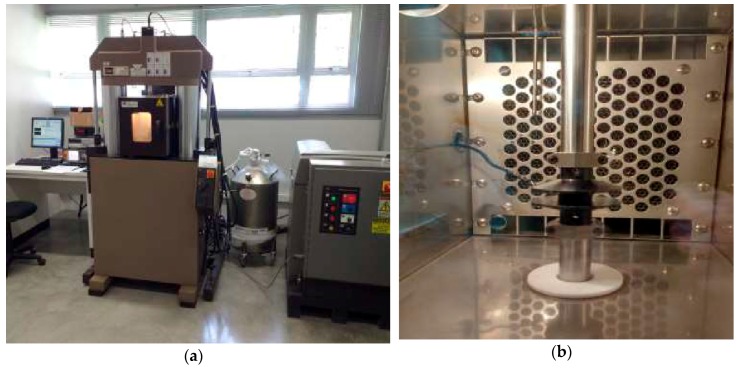
(**a**) Machine for universal tests MTS 831.50; (**b**) inside view of its thermal chamber.

**Figure 2 materials-12-01962-f002:**
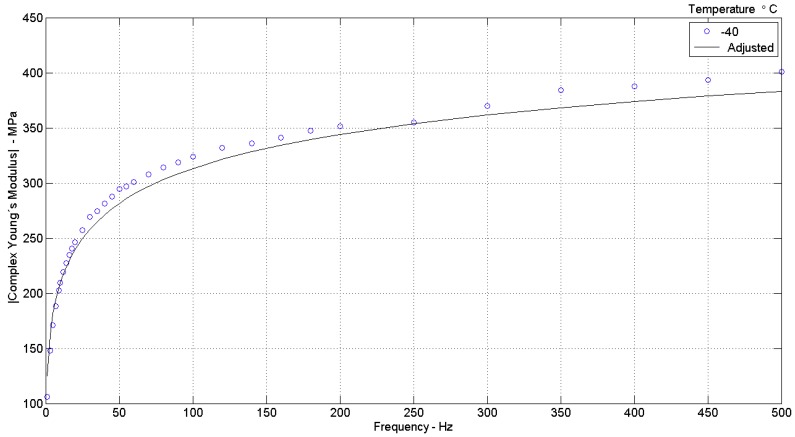
Variation of Complex Young’s modulus (experimental data “o”) for (adjusted curve by model parameters “−”), T = −40 °C, ε = −0.05 (λ = 0.95) [29].

**Figure 3 materials-12-01962-f003:**
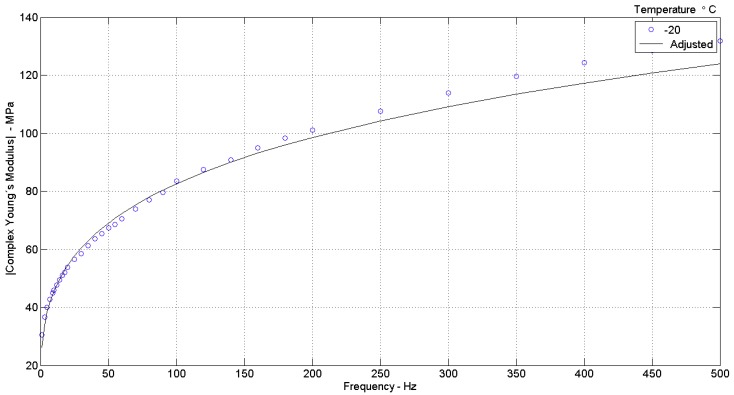
Variation of Complex Young’s modulus (experimental data “o”) for (adjusted curve by model parameters “−”), T = −20 °C, ε = −0.07 (λ = 0.95) [29].

**Figure 4 materials-12-01962-f004:**
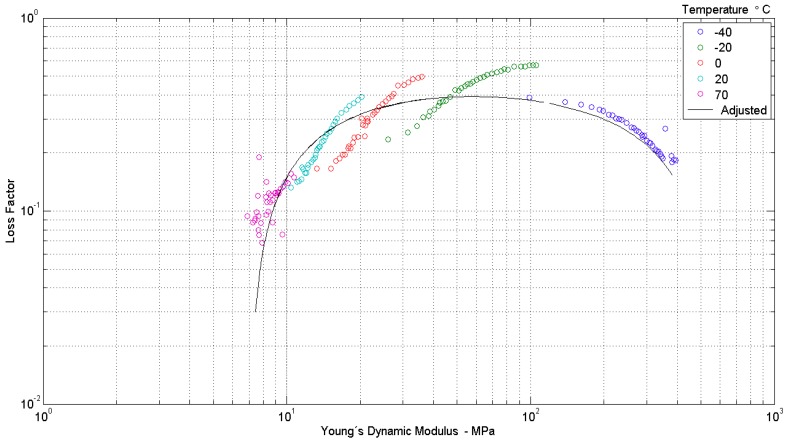
Wicket plot for the sample data at different temperatures (“o”) and adjusted curve by model parameters (“−”); (λ = 0.95) [29].

**Figure 5 materials-12-01962-f005:**
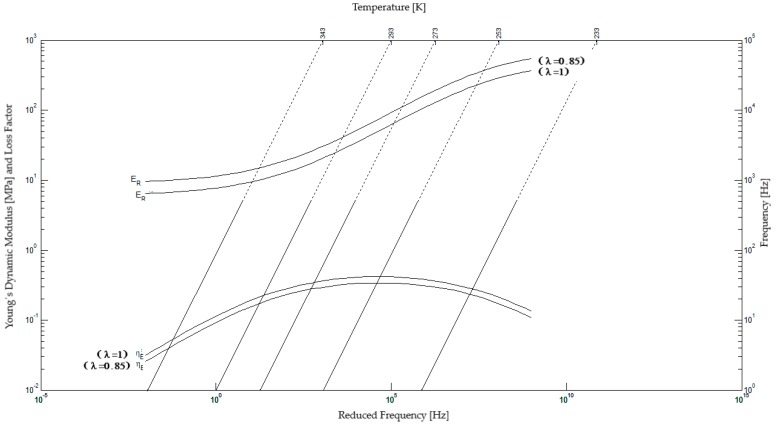
Reduced-frequency nomogram for Butyl Rubber for preloads of 0% and 15% of longitudinal strain [29].

**Figure 6 materials-12-01962-f006:**
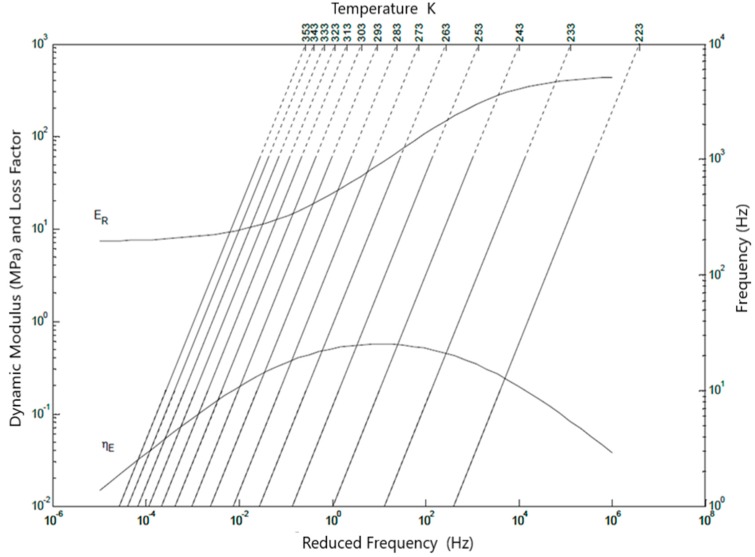
Reduced frequency nomogram for Butyl Rubber. *E*_0_ = 7.21 × 10^6^ Pa, $E_{\infty }$ = 4.57 × 10^6^ Pa, β = 0.417, $b_{0}$ = 0.0223 and $T_{0}$ = 243 K.

**Table 1 materials-12-01962-t001:** Performance conditions for tests.

Temperatures (°C)	−50, −40, −20, 0, 20 and 70
Strain Mode	Compression
Average Strains	−1%, −3%, −5%, −7%, −10%, −12% and −15%
Amplitude of harmonic excitation(mm)	+/−0.01
Excitation Frequencies (Hz)	1, 3, 5, 7, 9, 12, 14, 16, 18, 20, 25, 30, 35, 40 45, 50, 55, 60, 70, 80, 90, 100, 120, 140, 160, 180, 200, 250, 300, 350, 400, 450, 500, 550, 600, 700, 800, 900, 1000

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
