# Peer review of "Integrated Dynamic Characterization of Thermorheologically Simple Viscoelastic Materials Accounting for Frequency, Temperature, and Preload Effects"

_materials, 2019, doi:10.3390/ma12121962_

Round 1
Reviewer 1 Report
This article presents how to use integrated dynamic characterization method in vibration insulation studies. Moreover, hybrid optimization methods is harnessed for experimental data fitting. Article is well done, including appropriate numerical and experimental results and considerations. There is some typing errors, e.g. line 61, 162, 189, 200, 220, 256, 270 and 279 which needs to be taken care of. Also literature review was a bit out-of-date, so please include in this article summary of recents advancement on the field. In overall, despite these minor issues that needs to be taken care of, I am willing to accept this article publishing.
Author Response
First of all, we would like to thank all the reviewers for their valuable comments, which have clearly improved the contents of our paper.
Obs.: In the article, text marked in yellow corresponds to changes indicated by this reviewer
# - Spelling errors pointed out in lines 61, 162, 189, 200, 220, 256, 270 and 279. All have been corrected.
Line 61: word suppressed ("is";) (suppressed by another Reviewer)
Line 162: letter corresponding to the corrected symbol (1 -> lambda); ";) (suppressed by another Reviewer)
Line 189: word corrected moduli -> modulus; ";) (suppressed by another Reviewer)
Line 200: letter corresponding to a corrected symbol (epsilon); (suppressed by another Reviewer)
Line 220: missing parentheses corrected; (suppressed by another Reviewer)
Line 256: corrected punctuation failure; (suppressed by another Reviewer)
Line 279: word deleted.
* Note:
# - Literature revision. It has been revised and more recent work has been introduced. The literature review was completely restructured. It was divided in a first part on the mathematical models applied for the dynamic identification of the viscoelastic materials and a second part on the use of numerical methodologies applied to the existing mathematical models. This change produced a complete revision of the bibliographic review due to the restructuring (change of order of appearance).
Reviewer 2 Report
Review of the article materials-485051-v1 “Integrated dynamic characterization method applied to thermorheologically simple viscoelastic materials subject to frequency, temperature and preload influence” by E. Olienick, E. Lopes and C. Bavastri.
The article described a hybrid procedure for the dynamic characterization of Mooney-Rivlin visco-hyperelastic materials. The identification process accounted for effects of frequency, temperature and preload.
The subject of the study is very interesting. However, major revisions are required before final publication of the article in the Materials journal. First, the results of the identification process should be discussed in more detail pointing out more critically differences with the literature. Second, the proposed model should be clarified also in view of the results obtained from the identification process. Third, style, grammar and notation should be improved.
Ø The title of the article should read “Integrated dynamic characterization of thermorheologically simple viscoelastic materials accounting for frequency, temperature and preload effects”.
Ø The title of Section 2.2 should read “Effect of temperature”.
Ø The title of Section 2.3 should read “Effect of static load”.
Ø The title of Section 5 should read “Discussion and Conclusions”. Furthermore, this section should be expanded by discussing results in more detail and presenting more extensive comparisons with literature.
Ø Recall the concept of “thermorheologically simple” material. Furthermore, clarify the sentence “The condition of the thermorheologically simple material is reflected in the form of the curve [11]” (Section 4, pag. 11, lines 313-314).
Ø Section 2.3, pag. 7. In Eqs. (35,36), the last term of the RHS “E(W)” should read “Q(W)” to be consistent with the symbolism of Eq. (34). While it is clear that E(W) generally refers to dynamic components, general readers may be confused by the changing notation.
Ø Clarify derivation of Eq. (38).
Ø Section 2.3, pag. 8. In Eqs. (41,42), “kpcC2” should read “kpcC1”.
Ø Please clarify the physical meaning of the negative value (-2.1930) indicated for kpc in Section 4, pag. 10, line 281. Furthermore, the identified value of the unknown parameter To was not indicated in lines 280-281, pag. 10 of Section 4.
Ø The reduced-frequency nomogram is not clear. How can we determine from this nomogram the error values mentioned in the last paragraph of Section 4 (pag. 13, lines 354-359)? Please note that absolute temperature must be denoted as “K” not “°K”.
Ø References should be listed more precisely. Please implement the following changes in the revised article.
- Refs. [1,9,10,17,23,24,25,26]. Please type book/theses/standard titles in Italics. Furthermore, use capital initial letters for each title word. For example, Ref. [1], “Applications of generalized derivatives to viscoelasticity” should read “Applications of Generalized Derivatives to Viscoelasticity”;
- Do not use the notation Volume for journal papers. For example, Ref. [2], “Volume 15(4)” should read “15(4)”;
Ref. [10], pag. 14, line 401. “U.S.A.” should read “USA”;
- Ref. [16], pag. 14, lines 411-412. “Caracterização Dinâmica Integrada de Elastômeros por Derivadas Generalizadas” should read “Caracterização dinâmica integrada de elastômeros por derivadas generalizadas”;
- Ref. [19], pag. 14, line 419. “Journal of AIAA” should read “AIAA journal”.
Ø Style and grammar should be improved. Please implement the following changes in the revised article checking their consistency with the original content of the article.
- Abstract, pag. 1, line 12. “The most of the currently” should read “Most of the currently”
- Abstract, pag. 1, lines 13-14 & 15. “an effect with influence on the dynamic... here of preload” should read “a parameter affecting dynamic... here as preload”
- Abstract, pag. 1, line 21. “applied to all the experimental data at one go” should read “applied to all experimental data at hand”
- Abstract, pag. 1, line 23. “of the methodology of this work and the traditional one” should read “of the present methodology and traditional approaches”
- Abstract, pag. 1, lines 24-25 & 26. “The ensuing characterization presents promising results... materials which are used” should read “The present results prove... materials used”
- Introduction, pag. 1, line 34. “Aiming at obtaining an accurate representation of the behavior” should read “In order to accurately represent behavior”
- Introduction, pag. 2, line 46. “allowing obtaining a single” should read “and then merged into a single”
- Introduction, pag. 2, lines 47 & 48. “approach for the characterization... is presented” should read “approach to characterization... was presented”
- Introduction, pag. 2, line 49. “annular proof-body” should read “annular specimen”
- Introduction, pag. 2, line 51. “present an” should read “presented an”
- Introduction, pag. 2, line 61. “Temperature is the... influence in” should read “Temperature has the... influence on”
- Introduction, pag. 2, line 67. “One deficiency regarding the” should read “A limitation of the”
- Introduction, pag. 2, line 88. “preload to the” should read “preload into the”
- Section 2.1, pag. 3, line 117. “Connsidering that” should read “Considering that”
- Section 2.1, pag. 4, line 130. “and as defined” should read “and b as defined”
- Section 2.1, pag. 4, line 133. “dynamics behavior” should read “dynamic behavior”
- Section 2.1, pag. 4, lines 135-136. “However, the only external... designs has been temperature” should read “However, temperature is the only external factor usually considered in vibration control design”
- Section 2.2, pag. 4, line 141. “obtaining a wide” should read “to obtain a comprehensive”
- Section 2.2, pag. 4, line 143. “dynamics properties” should read “dynamic properties”
- Section 2.2, pag. 4, lines 148-149. “is the equation known as WLF, or Williams-Landel-Ferry” should read “is the Williams-Landel-Ferry (WLF) model”
- Section 2.2, pag. 4, lines 151-152. “temperature of the experimental data” should read “temperature at which experimental data are collected”
- Section 2.2, pag. 4, line 153. “The use of” should read “Substituting”
- Section 2.3, pag. 5, line 157. “is found in” should read “is given in”
- Section 2.3, pag. 5, lines 158 & 159. “is used by... section 2.1” should read “was used in... Section 2.1”
- Section 2.3, pag. 5, lines 162-163. “The equation of the energy density function for an incompressible material, said Mooney-Rivlin material” should read “In the Mooney-Rivlin model, the strain energy density function W for an incompressible material”
- Section 2.3, pag. 5, line 164. “where W is the energy density function of strain, C1” should read “where C1”
- Section 2.3, pag. 5, line 166. “Cauchy-Green strain tensor to the left” should read “left Cauchy-Green strain tensor”
- Section 2.3, pag. 5, lines 166-167. “The Cauchy-Green strain tensor to the left” should read “The B tensor”
- Section 2.3, pag. 5, line 169. “When hyperelastic materials are regarded as incompressible” should read “For incompressibile hyperelastic materials”
- Section 2.3, pag. 7, line 196. “see Eq. (5)... Eq. (33) may” should read “see Eqs. (5-7)... Eq. (31) may”
- Section 2.3, pag. 7, line 200. “Considering that is” should read “Considering that e is”
- Section 2.3, pag. 8, lines 203-204. “As presented in this” should read “In this”
- Section 2.3, pag. 8, lines 210-212. “for they are based... in the present work.” should read “as they have the same theoretical foundations. However, they have been adapted to fit in the context of the present study”
- Section 3.2, pag. 8, lines 220 & 221. “(BT-806 55 was... proof-bodies” should read “(BT-806 55) was... specimens”
- Section 3.2, pag. 8, line 222. “proof-bodies” should read “specimens”
- Section 3.2, pag. 9, line 227. “proof-body” should read “specimen”
- Caption of Figure 1, pag. 9, line 234. “Proof-body;” should read “Specimen.”
- Title of Table 1, pag. 9, line 236. “Tabela” should read “Table 1”
- Section 3.3, pag. 9, lines 245 & 246. “those parameters that compose the models representing the influences... obtained through” should read “the other parameters describing effects... obtained from”
- Section 3.3, pag. 9, line 248. “allows establishing the initial parameters to be applied” should read “allows to set initial values of unknown material parameters used”
- Section 3.3, pag. 10, line 259. “process in terms of processing time” should read “process”
- Section 3.3, pag. 10, lines 261 & 262-263. “eq. (47)... can have a considerable reduction in the processing time” should read “Eq. (47)... may be considerably speeded up”
- Section 3.3, pag. 10, line 263. “performed” should read “done”
- Section 3.3, pag. 10, lines 267-268. “and lower and upper... parameters were not applied” should read “and bounds on model parameters were not considered”
- Section 4, pag. 10, lines 278 & 279. “maximal number of interactions allowed... stop tolerance” should read “limit number of iterations... convergence limit”
- Section 4, pag. 10, line 281. “=0.3166, 1=7.1119, 2=132.72” should read “b=0.3166, q1=7.1119, q2=132.72”
- Section 4, pag. 11, line 287. “It is recall” should read “It should be noted”
- Caption of Figure 2, pag. 11, lines 295 & 296. “Curves of the modulus of Complex... x” should Curves of the modulus of Complex... x (regenerated” should read “Variation of Complex... for (adjusted”. Please remove plot title.
- Caption of Figure 3, pag. 11, lines 311 & 312. “Curves of the modulus of Complex... x (regenerated” should read “Variation of Complex... for (adjusted”. Please remove plot title.
- Section 4, pag. 11, line 313. The sentence “The Wicket plots show the adjustment from the point of view of the used mathematical” should be rewritten more clearly.
- Section 4, pag. 11, line 315. “is a typical representant of the wicket plots for the 5 preload conditions” should read “is a typical representation of the Wicket plots for five preload levels”
- Caption of Figure 4, line 334. “and regenerated curve” should read “and adjusted curve”
- Section 4, pag. 12, line 350. “elastomeric devices” should read “elastomeric devices.”
- Section 4, pag. 12, lines 351-352. “the methodology of model parameters identification” should read “the proposed parameter identification methodology”
- Section 4, pag. 13, line 355. “preload with a 15% strain by compression in relation to the initial length” should read “a 15% compressive strain preload”
- Section 5, pag. 13, line 365. “of other influences to those” should read “of other effects besides those”
- Section 5, pag. 13, line 368. “of the in the” should read “in the”
Author Response
First of all, we would like to thank all the reviewers for their valuable comments, which have clearly improved the contents of our paper.
First of all, we would like to thank all the reviewers for their valuable comments, which have clearly improved the contents of our paper.
Obs.: In the article, text marked in green corresponds to changes indicated by this reviewer
# - Change in Title: accepted. The new title is: “Integrated dynamic characterization of thermoreologically simple viscoelastic materials accounting for effects of frequency, temperature and preload”.
# - Section 2.2: accepted. The new title is: “Effect of temperature”
# - Section 2.3: accepted. The new title is: “Effect of static load”
# - Section 5 Title: Partially Accepted. A new section was created for Conclusions (item 6) and Section 5 was renamed Discussion (item 5). In this chapter the results of the identification of the same material (BT806-55) were inserted only with the consideration of temperature and frequency, using the same technique of this work and for a better understanding of the comprehensiveness of the results, an appendix was created with the corresponding nomograms five deformations due to the preloads.
# - Section 4: Concept of the term ‘Termoreologically simple viscoelastic material’: For a better understanding of the term, in the context of this work and allusion to the shape of the wicket plot curve obtained for the experimental data of this work, a paragraph extracted from Jones, DIG, "Handbook of Viscoelastic Vibration Damping" reference 14.
In line 329, page 12 was introduced:
“a plot of the loss factor versus the real part of the complex Young’s modulus, called Wicket plot. For a thermorheologically simple material, the complex modulus data points should cluster around an unique inverted-U shaped curve in a Wicket plot [33]. A viscoelastic material is regarded as thermorheologically simple if the frequency-temperature superposition principle holds, that is, if the frequency and temperature dependencies can be worked out in such a way that a single dependence on a compound variable, combining the effects of both frequency and temperature, results [13]. This compound variable is known as reduced frequency.”
# - Section 2.3: Error in equation - E(W) por Q(W) (Eqs. (35) and (36)). The equation was corrected. The misuse term E (W) was replaced for Q (W).
# - Clarification of equation (38). In order to obtain the equations of E (W, T, lambda) and eta (W, T, lambda) - eqs (37) and (38) - we lacked the condition that with lambda = 1, eta 1 = 0. This definition can be seen in Nashif et al. (ref. 13), "Vibration Damping". The absence of this condition was corrected.
# - Seção 2.3 Kpc. C2 (Kpc multiplying incorrect parameter). Equations (41) and (42) were corrected. In the last version, Kpc multiplies the term C1.
# - Kpc negative sign in results. The authors agree with the reviewer and believe that the negative signal for this parameter kpc is due to the mathematical model used in this work (ref. [13]). Obviously, C1 e C2 are, in general, positive.
In line 316 – page 12 was introduced the following text:
It is observed that the constants C2 and C1 of the Mooney Rivlin model, the ratio of which is equal to kpc, are relative to the material and their values are obtained from experimental data. However, in the current optimization procedure, the concern lies in their ratio, kpc, and not in their separate values.
# - Nomogram - comparison. Section 4. The temperature scale was corrected [K]. The error is obtained through comparisons of the values obtained for modulus and loss factor, for the condition where lambda = 1 (not considering the preload) and lambda assumes the value 0.85 (15% of longitudinal deformation due preload). Note that these two curves are available in the same nomogram. In the Nomograms available by the manufacturer of the viscoelastic material, lambda is always equal to 1, that is, the deformation is not considered. Thus, by comparing for the same frequency and temperature, the values obtained from the curves, it is possible to obtain a relation between the values normally obtained by the commercial nomograms and with the consideration of the preload.
# - Correction in Bibliographic References.
- Ref[1, 9, 10, 17, 23, 24, 25 e 26] * - corrected - the denomination of the work is now in capital initial letters.
(*) – These articles now have new reference numbers, due to the corrections requested in the "Introduction" section by another reviewer.
- Volume notation for newspaper papers has been corrected. Now it is in the form "xx (y)"
- Denomination of United States of America became USA.
- Size of the initial characters of the words that make up the thesis title, with the exception of the first, must be in lowercase. adjusted
- Title of the AIAA Journal corrected for "AIAA Journal"
# - Style and Grammar
Abstract
All requested changes have been implemented:
- Abstract pag. 1, line 12: (“Most of the currently” replaced “The most of the currently”.
- Abstract pag. 1, lines 13-14: “a parameter affecting dynamic…. here as preload” replaced “an effect with influence on the dynamic…here of preload”, as can be seen in line 13 – page 1.
- Abstract pag. 1, line 21: “applied to all experimental data at hand” replaced “applied to all the experimental data at one go”.
- Abstract pag. 1, line 23: “of the present methodology and traditional approaches” replaced “of the methodology of this work and the traditional one”.
- Abstract pag. 1, line 24-25 and 26: “The present results prove… materials used” replaced “The ensuing characterization presents promising results…materials which are used”.
Introduction
- “ Introduction pag. 1, line 34: In order to accurately represent behavior” replaced “Aiming at obtaining an accurate representation of the behavior”. Suppressed by another revisor.
- “and then merged into a single” replaced “allowing obtaining a single”. Suppressed by another revisor.
- “approach to characterization... was presented” replaced “approach for the characterization... is presented” . Suppressed by another revisor.
- “annular specimen” replaced “annular proof-body” . Suppressed by another revisor.
- “presented an” replaced “present an”. Suppressed by another revisor.
- “Temperature has the... influence on” replaced “Temperature is the... influence in”.
Suppressed by another revisor.
- “A limitation of the” replaced “One deficiency regarding the”. Suppressed by another revisor.
- “preload into the” replaced “preload to the”. Suppressed by another revisor.
Section 2.1
- Section 2.1, pag. 3, line 117 (152);“Considering that” replaced “Connsidering that”.
- Section 2.1, pag. 4, line 117 (165): “and b as defined” replaced “and as defined”.
- Section 2.1, pag. 4, line 133 (169) “dynamic behavior” replaced “dynamics behavior”.
- Section 2.1, pag. 4, line 135-136 (170-171) “However, temperature is the only external factor usually considered in vibration control design” replaced “However, the only external... designs has been temperature”.
Section 2.2
- Section 2.2, pag 4, line 141(177):“to obtain a comprehensive” replaced “obtaining a wide”.
- Section 2.2, pag 4, line 143(179): “dynamic properties” replaced “dynamics properties” .
- Section 2.2, pag 4, line 148-149(183-184): “is the Williams-Landel-Ferry (WLF) model” replaced “is the equation known as WLF, or Williams-Landel-Ferry” should read “.
- Section 2.2, pag 4, line 151-152(188): “temperature at which experimental data are collected” replaced “temperature of the experimental data”.
- Section 2.2, pag 4, line 153(189): “Substituting” replaced for “The use of”.
Section 2.3
- Section 2.3, pag. 5, line 157(193):is given in” replaced “is found in”.
- Section 2.3, pag. 5, line 158-159(198-199): “was used in... Section 2.1” replaced “is used by... section 2.1”.
- Section 2.3, pag. 5, line 162-163(198-199): “In the Mooney-Rivlin model, the strain energy density function W for an incompressible material” replaced “The equation of the energy density function for an incompressible material, said Mooney-Rivlin material”.
- Section 2.3, pag. 5, line 164(200):“ “where C1” replaced “where W is the energy density function of strain, C1” .
- Section 2.3, pag. 5, line 166(202):“ “left Cauchy-Green strain tensor” replaced “Cauchy-Green strain tensor to the left”.
- Section 2.3, pag. 5, line 166-167(202):“ “The B tensor” replaced “The Cauchy-Green strain tensor to the left”.
- Section 2.3, pag. 5, line 169(204):“ “For incompressibile hyperelastic materials” replaced “When hyperelastic materials are regarded as incompressible”.
- Section 2.3, pag. 7, line 196(231):“ “see Eqs. (5-7)... Eq. (31) may” replaced “see Eq. (5)... Eq. (33) may”.
- Section 2.3, pag. 7, line 157(235):“ “Considering that e is” replaced “Considering that is”.
- Section 2.3, pag. 8, line 203-204(242):“ “In this” replaced “As presented in this”.
- Section 2.3, pag. 8, line 210-212(248):“ “as they have the same theoretical foundations. However, they have been adapted to fit in the context of the present study” replaced “for they are based... in the present work.”.
Section 3.2
- Section 3.2, pag. 8, line 220-221(258):“ “(BT-806 55) was... specimens” replaced “(BT-806 55 was... proof-bodies”.
- Section 3.2, pag. 8, line 222(260) “specimens” replaced “proof-bodies”.
- Section 3.2, pag. 9, line 227(265) “specimen” replaced “proof-body”.
- In the caption of Figure 1, pag. 9: “specimen” replaced “Proof-body;”. Suppressed by another reviwer.
- In the Title of Table 1, pag. 9, line 236(273) “Table 1” replaced “Tabela”.
Section 3.3
- Section 3.3, pag. 9, lines 245-246(282): “the other parameters describing effects... obtained from” replaced “those parameters that compose the models representing the influences... obtained through”.
- Section 3.3, pag. 9, line 248(285): “allows to set initial values of unknown material parameters used” replaced “allows establishing the initial parameters to be applied” .
- Section 3.3, pag. 10, line 259(299): “process” replaced “process in terms of processing time”.
- Section 3.3, pag. 10, line 263(300): “Eq. (47)... may be considerably speeded up” replaced “eq. (47)... can have a considerable reduction in the processing time”.
- Section 3.3, pag. 10, line 263(300): “done” replaced “performed”.
- Section 3.3, pag. 10, lines 267-268(282): “and bounds on model parameters were not considered” replaced “and lower and upper... parameters were not applied” .
Section 4
- Section 4, pag. 10, lines 278-279(314): “limit number of iterations... convergence limit” replaced “maximal number of interactions allowed... stop tolerance”.
- Section 4, pag. 10, line 281(316): “b=0.3166, q1=7.1119, q2=132.72” replaced b =0.3166, q1=7.1119, q2=132.72” .
- Section 4, pag. 11, line 287(326): “It should be noted” replaced “It is recall”.
- In the Caption of Figure 2, pag. 11, lines 295-296(327): “Variation of Complex... for (adjusted” replaced Curves of the modulus of Complex... x” should Curves of the modulus of Complex... x (regenerated” . The title was removed.
- In the Caption of Figure 3, pag. 11, lines 311-312(341): “Variation of Complex... for (adjusted” replaced “Curves of the modulus of Complex... x (regenerated. Removed by another reviewer.
-
- Section 4, pag. 11, line 313: The sentence “The Wicket plots show the adjustment from the point of view of the used mathematical” received an observation in parentheses.
- Section 4, pag. 11, line 315(336): “is a typical representation of the Wicket plots for five preload levels” replaced “is a typical representant of the wicket plots for the 5 preload conditions”.
- Section 4, pag. 11, line 334(359): “and adjusted curve” replaced “and regenerated curve”.
- Section 4, pag. 12, line 350(352): “elastomeric devices.” Replaced “elastomeric devices”.
- Section 4, pag. 12, lines 351-352(353): “the proposed parameter identification methodology” replaced “the methodology of model parameters identification”.
- Section 4, pag. 13, line 355(367): “a 15% compressive strain preload” replaced “preload with a 15% strain by compression in relation to the initial length” .
Section 5
- Section 5, pag. 13, line 365(421): “of other effects besides those” replaced “of other influences to those”.
- Section 5, pag. 13, line 368(422): “in the” replaced “of the in the”.
Reviewer 3 Report
In this paper, the mechanical properties of viscoelastic materials are investigated. A dynamic characterization is developed by taking into account the effect of temperature, frequency and static preload. In particular, it is proven that the preload effect is relevant in vibration insulation design.
The present paper is well written and clearly arranged. The Introduction is very detailed. The mathematical model of dynamic characterization is presented in a rigorous manner. In particular, the section on the static load influence in the framework of the Mooner-Rivlin hyper-elastic theory is properly treated. The experimental results are clearly reported and discussed.
However, there are some important aspects to be carefully taken into account by the Authors.
1) There are several English Language mistakes within the text that should be corrected.
2) The quality of the figures should be deeply improved (the curves are not clear).
3) The figures related with the experimental set-up should be modified (by increasing Figure 1a and Figure 1b, by eliminating Figure 1c and inserting a new figure on the microstructure of the viscoelastic material).
4) A relevant shortcoming of this paper concerns the applications of the analyses performed in the vibration insulation design. For example, one of the most relevant properties of viscoelastic materials is given by their very high damping contribution in composite structures, e.g., sandwich beams or multi-layer shells. Therefore, in the Introduction, it should be emphasized the application of viscoelastic materials as damping materials in the vibration insulation design and the following relevant papers should be added and discusses.
a) Catania, G.; Strozzi, M. Damping oriented design of thin-walled mechanical components by means of multi-layer coating technology. Coatings 2018, 8, 73.
b) Yu, L.; Ma, Y.; Zhou, C.; Xu, H. Damping efficiency of the coating structure. International Journal of Solids and Structures 2005, 42, 3045-3058.
c) Rongong, J.A.; Goruppa, A.A.; Buravalla, V.R.; Tomlinson, G.R.; Jones, F.R. Plasma
deposition of constrained layer damping coatings. Proceedings of the Institution of Mechanical Engineers, Part C: Journal of Mechanical Engineering Science 2004, 218, 669-680.
Therefore, by considering the previously reported observations, in the opinion of the Reviewer this paper should be accepted for publication by minor revision.
Author Response
First of all, we would like to thank all the reviewers for their valuable comments, which have clearly improved the contents of our paper.
1) English Languages Mistakes: Accepted request. Several terms and excerpts have been modified.
2) Quality of Figures: The image files (Figures 1, 2, 3, 4, 5, 6 were replaced by a more precise pattern, in the case of ".png".) Attached figures in Appendix A were also inserted with the same pattern.
3) Figure 1: Fig. 1c was removed, as requested. However, the suggestion of insertion of a new figure with the microstructure of the viscoelastic material was not accepted, since it is not part of the scope of this work.
4) Suggestion for Insertion of Additional References: It was considered by the authors that these new references are not correlated to this work. New references were added in the bibliographic review, which are better related to this work.
Round 2
Reviewer 2 Report
Review of the article materials-485051-v2 “Integrated dynamic characterization of thermorheologically simple viscoelastic materials accounting for frequency, temperature and preload effects” by E. Olienick, E. Lopes and C. Bavastri.
The authors have taken care of the comments made by this reviewer and improved the quality of their article, which can now be published in the Materials journal with the following amendments.
Ø Section 4, pag. 13, lines 364-366 & pag. 14, lines 370-375. The paragraphs “A plot of the loss... a Wicket plot [33]” and “A viscoelastic material is… five preload levels” are repeated twice and should be deleted. Please do not show revisions/insertions/deletions in the submitted manuscript, just show the text highlighted where necessary.
Ø The identified value of the unknown parameter To is not indicated in lines 342-343, pag. 12 of Section 4.
Ø References should be listed more precisely. Please implement the following changes in the revised article.
- Do not use “pp” before page numbers of journal articles;
- Refs. [2] and [28] are the same. Please remove [28] and renumber references accordingly;
- Refs. [2,3]. Quote publication year as “1983” not “1983a” and “1983b”;
- Refs. [3,4], pag. 21. “AIAA journal” should read “AIAA Journal”;
- Ref. [7], pag. 21. “anelasticity” should read “Anelasticity”;
- Ref. [15], pag. 21. Clarify the notation “3o”;
- Ref. [18], pag. 21. “Computational materials” should read “Computational Materials”;
- Ref. [19], pag. 21. Type publication year in bold.
Ø Style and grammar needs further improvement.
- Introduction, pag. 2, line 89. “data is dealt” should read “data are dealt”
- Introduction, pag. 2, line 94. “through its transmissibility” should read “through the transmissibility”
- Introduction, pag. 3, line 104. “solution an inverse” should read “solution of an inverse”
- Introduction, pag. 3, line 111. “and the temperature” should read “and temperature”
- Introduction, pag. 3, line 143. “joining a genetic” should read “combining a genetic”
- Section 2.3, pag. 9, line 257. “may be written” should read “be written”
- Section 3.3, pag. 12, lines 322 & 324. “eq. (47)... may be considerable speed up in the process” should read “Eq. (47)... may become much faster”
- Section 4, pag. 12, line 340. “interactions allowed” should read “iterations”
- Figure 2, pag. 13, below line 352. “Temperatura °C” should read “Temperature °C”
- Figure 3, pag. 14. “Temperatura °C” should read “Temperature °C”
- Section 4, pag. 14, line 374. “representative” should read “portray”
- Section 5, pag. 15, line 395. “may reach up” should read “errors may reach up”
- Section 5, pag. 16, line 410. “The curves correspondent to dynamic modulus and los factor... in figure” should read “The curves correspondent to dynamic modulus and loss factor... in Figures”
- Section 5, pag. 16, lines 411 & 412. “Another curves... los factor, in figure 5, corresponding... preload compression” should read “The other curves... loss factor, in Figure 5, correspond... compression preload”
- Section 5, pag. 16, line 420. “plotted at the same” should read “plotted in the same”
- Title of left vertical axis of Figure 6, pag. 17. “Lost Factor” should read “Loss Factor”
Author Response
Author's Notes to Reviewer:
All The requested corrections were implemented:
Ø Section 4, pag. 13, lines 364-366 & pag. 14, lines 370-375. The paragraphs “A plot of the loss... a Wicket plot [33]” and “A viscoelastic material is… five preload levels” are repeated twice and should be deleted. Please do not show revisions/insertions/deletions in the submitted manuscript, just show the text highlighted where necessary.
R.: Both paragraphs have been deleted.
Ø The identified value of the unknown parameter To is not indicated in lines 342-343, pag. 12 of Section 4.
R.: was inserid , on page 342
Ø References
- Do not use “pp” before page numbers of journal articles; R.: done.
- Refs. [2] and [28] are the same. Please remove [28] and renumber references accordingly; R.: done.
- Refs. [2,3]. Quote publication year as “1983” not “1983a” and “1983b”; R.: adjusted.
- Refs. [3,4], pag. 21. “AIAA journal” should read “AIAA Journal”; R.: adjusted.
- Ref. [7], pag. 21. “anelasticity” should read “Anelasticity”; R.: adjusted
- Ref. [15], pag. 21. Clarify the notation “3o”; R.: changed to “3rd”
- Ref. [18], pag. 21. “Computational materials” should read “Computational Materials”; R.: done.
- Ref. [19], pag. 21. Type publication year in bold. R.: done
Ø Style and gramar:
- Introduction, pag. 2, line 89. “data is dealt” should read “data are dealt” R.: adjusted.
- Introduction, pag. 2, line 94. “through its transmissibility” should read “through the transmissibility” R.: adjusted
- Introduction, pag. 3, line 104. “solution an inverse” should read “solution of an inverse” R.: adjusted.
- Introduction, pag. 3, line 111. “and the temperature” should read “and temperature” R.: adjusted.
- Introduction, pag. 3, line 143. “joining a genetic” should read “combining a genetic” R.: adjusted.
- Section 2.3, pag. 9, line 257. “may be written” should read “be written” R.: adjusted.
- Section 3.3, pag. 12, lines 322 & 324. “eq. (47)... may be considerable speed up in the process” should read “Eq. (47)... may become much faster” R.: adjusted.
- Section 4, pag. 12, line 340. “interactions allowed” should read “iterations” R.: adjusted
- Figure 2, pag. 13, below line 352. “Temperatura °C” should read “Temperature °C” R.: adjusted.
- Figure 3, pag. 14. “Temperatura °C” should read “Temperature °C” R.: adjusted.
- Section 4, pag. 14, line 374. “representative” should read “portray” R.: adjusted
- Section 5, pag. 15, line 395. “may reach up” should read “errors may reach up” R.: adjusted.
- Section 5, pag. 16, line 410. “The curves correspondent to dynamic modulus and los factor... in figure” should read “The curves correspondent to dynamic modulus and loss factor... in Figures” R.: adjusted.
- Section 5, pag. 16, lines 411 & 412. “Another curves... los factor, in figure 5, corresponding... preload compression” should read “The other curves... loss factor, in Figure 5, correspond... compression preload” R.: adjusted.
- Section 5, pag. 16, line 420. “plotted at the same” should read “plotted in the same” R.: adjusted.
- Title of left vertical axis of Figure 6, pag. 17. “Lost Factor” should read “Loss Factor” R.: adjusted